# SDA-CoT: Structure-driven Dynamic Active Chain-of-Thought

## Abstract

Chain-of-Thought (CoT) improves large language model (LLM) reasoning by making intermediate steps explicit. Many methods still rely on fixed human exemplars and ignore structural cues such as entity relations, which can lead to confident but faulty reasoning paths. This paper presents Structure-driven Dynamic Active Chain-of-Thought (SDA-CoT), a framework that combines uncertainty-based exemplar selection with structure-aware reasoning to address these issues. SDA-CoT uses Bayesian Active Learning (BAL) to select exemplars with high uncertainty and strong expected value, applies entity and relation extraction to build relational structures, and then produces reasoning paths that remain logically coherent and consistent with context. Across three dataset families (commonsense reasoning, logical reasoning, and math word problems) and two LLMs (LLaMA2-13B and DeepSeek-R1), SDA-CoT surpasses standard CoT methods. In LLaMA2-13B, the accuracy increases by 9% on average, with a gain of 12% on GSM8K. In DeepSeek-R1, the average gain is 8%. The combination of entity–relation analysis with adaptive prompting produces robust and interpretable CoT and provides the first empirical evidence that BAL can significantly improve CoT reasoning in LLMs.

## 1 Introduction

Large language models (LLMs) (Brown et al., 2020; Chowdhery et al., 2023; OpenAI, 2023) have changed natural language processing (NLP) by showing strong in-context learning on tasks such as question answering, arithmetic reasoning, and commonsense inference. One widely used strategy for complex problem solving is Chain-of-Thought (CoT) prompting (Wei et al., 2022b). When a prompt asks the model to break a problem into explicit steps, CoT makes the reasoning trace visible and often improves prediction accuracy.

Recent studies find that CoT behavior depends on the phrasing of exemplars. As shown in Figure 1, even a simple task that mixes arithmetic and proportion can shift course after small edits to the preceding exemplars. With an ill-suited exemplar (Prompt 1), the model performs a spurious computation and wrongly concludes that John lost to five opponents by misusing percentage arithmetic. With a well-structured exemplar (Prompt 2) that gives consistent logic and clear structure, the model reaches the correct answer of four opponents. These observations point to two challenges: exemplar choice governs the faithfulness of the reasoning process, and fixed prompts lack the flexibility needed to match task uncertainty and structural fit.

Despite broad interest in CoT, common practice has clear limits. Many methods use fixed, human-written exemplars (Wei et al., 2022b; Zhou et al., 2023) or template prompts that do not adapt across tasks. Such designs neglect task-specific uncertainty, often producing confident but erroneous reasoning chains, as shown in Figure 1. They also ignore structural cues in text, including entity interactions and relational dependencies, which matter for multi-hop and logical reasoning (Liu et al., 2024). Fixed exemplars transfer poorly across tasks that vary in difficulty or required domain knowledge. As a result, LLMs remain brittle in settings that call for dynamic, context-aware reasoning paths.

In this work, we introduce Structure-driven Dynamic Active Chain-of-Thought (SDA-CoT), a framework that integrates structural reasoning and dynamic prompting into a unified architecture

to address these limitations. First, it represents structural information, such as entity types, relations, and exemplars dependencies, as a latent graph, so the model can organize its reasoning in a coherent way. Second, it adapts its reasoning path by actively selecting high-quality exemplars that are strongly associated with high model uncertainty. In this way, SDA-CoT avoids spurious chains induced by mismatched prompts and produces task-aware, logically consistent reasoning trajectories.

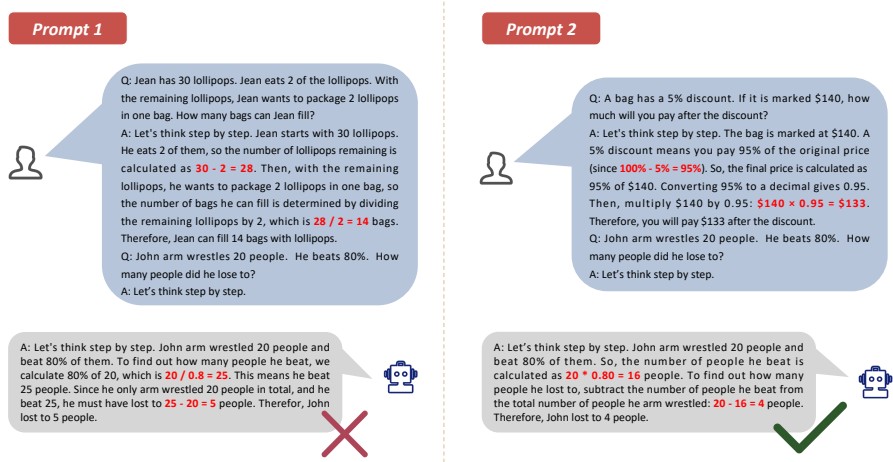

Figure 1: Example of the CoT prompting. The prompt influences how LLMs reach the final answer.

Specifically, SDA-CoT consists of three modules: (i) a structure extractor that builds relational representations by analyzing entity interactions and implicit dependencies; (ii) a dynamic reasoning planner that evaluates uncertainty across multiple CoT paths and expands the most informative ones; and (iii) a CoT generator that combines structural cues with uncertainty-aware exemplars to produce task-adaptive reasoning chains. Unlike prior methods that treat structural analysis and CoT as separate components, SDA-CoT couples them, yielding context-sensitive reasoning with stable logic and greater adaptability.

The contributions of this work are summarized as follows:

- SDA-CoT integrates entity and relation analysis with dynamic exemplar selection, enabling structure-aware CoT rather than relying on fixed templates. This design reduces reasoning errors caused by prompt mismatches and enhances the ability of LLMs to handle complex logical reasoning tasks.

- This is the first work to demonstrate that Bayesian active learning enhances CoT reasoning in LLMs. Our composite acquisition rule for exemplar selection combines disagreement, entropy, and confidence variance, targeting highly uncertain cases and providing annotation chains that calibrate reasoning across tasks.

- Comprehensive experiments on six benchmarks (StrategyQA, CSQA, LogiQA, HotpotQA, 2WikiMultiHopQA, GSM8K) with LLaMA2-13B and DeepSeek-R1 show consistent gains over strong CoT variants, with an average improvement of 7.12%.

- CoT increases model stability and interpretability and produces more accurate and reliable reasoning paths in domains that require multi-hop inference and arithmetic reasoning.

## 2 RELATED WORK

### 2.1 CHAIN OF THOUGHT

Chain-of-Thought (CoT) prompting has become a standard approach for LLMs to handle complex reasoning through intermediate steps that lead to a final answer (Wei et al., 2022b; Kojima et al., 2022). Early studies show that even a short prefix such as *"Let's think step by step"* can elicit multi-hop inference without additional training (Kojima et al., 2022). To increase stability and robustness,

several methods have been proposed. Auto-CoT (Zhang et al., 2023b) constructs exemplars automatically, and Complex-CoT (Fu et al., 2023b) adjusts the depth of reasoning to match task difficulty.

Most methods still rely on fixed exemplars or prompt templates, which limits adaptation to new tasks or shifts in input distribution. Conventional CoT approaches also pay little attention to structural cues such as entity dependencies and relational hierarchies, which matter for logical and multi-hop reasoning. This work combines CoT prompting with structure-aware entity–relation modeling and active prompt adaptation, and it produces reasoning chains that are more targeted and easier to interpret.

## 2.2 BAYESIAN ACTIVE LEARNING

Active Learning (AL) reduces annotation cost by iteratively choosing informative samples for labeling (Li et al., 2025). Conventional AL often uses simple confidence scores that do not capture uncertainty under scarce data or parameter ambiguity. In many settings, uncertainty measures such as entropy or disagreement are treated separately (Xu & Zhang, 2024; Li et al., 2025), and the result is weaker sample ranking. The effect is acute for CoT exemplar selection: ambiguous reasoning paths or conflicting intermediate steps can lower LLM performance.

Bayesian Active Learning (BAL) (Houlsby et al., 2011b) models a posterior over predictions and treats uncertainty in a unified way. It separates aleatoric noise from epistemic uncertainty (Kendall & Gal, 2017), and this split directs selection toward samples that are both ambiguous and informative. Rather than relying on point scores, BAL estimates predictive distributions; prior work reports gains in image recognition (Gal et al., 2017), neural machine translation (Cheng et al., 2025), and text classification (Ash et al., 2020), with better sample efficiency than heuristic AL. The same approach suits conditions with few labels, distribution shift, or multi-step inference, such as CoT reasoning with LLMs. In our use, BAL supports an uncertainty-driven strategy for exemplar choice in CoT prompting. We combine disagreement, entropy, and confidence variance to capture complementary views of predictive uncertainty—variance is measured across stochastic passes or model variants—and we pick the hardest and most informative pool items. We then add manual chains for these items to the exemplar set, which lowers ambiguity in the model's reasoning and improves stability across varied inputs.

## 2.3 NAMED ENTITY RECOGNITION & RELATION EXTRACTION

Named Entity Recognition (NER) and Relation Extraction (RE) are core tasks in information extraction; they identify entities and classify their relations in unstructured text (Detroja et al., 2023). With LLMs, NER has moved from sequence labeling to generation-based paradigms (Wang et al., 2025). This shift allows models to adapt to diverse domains. RE has also been cast as a question answering task (Efeoglu & Paschke, 2024), with prompt-based methods that work in zero-shot and few-shot settings.

Recent studies have emphasized the importance of leveraging entity and relation structures to enhance reasoning. For example, ERA-CoT (Liu et al., 2024) adds explicit and implicit relational triplets to the CoT pipeline; this design helps models separate entity semantics and reason in a more grounded way. Many approaches still treat structure extraction and CoT prompting as separate steps. SDA-CoT links NER and RE with dynamic reasoning so that the reasoning path is refined in response to relational cues and exemplar uncertainty. The result is higher accuracy and clearer reasoning in tasks that require multi-hop inference.

## 3 METHOD

The SDA-CoT framework enhances multi-step reasoning by combining structural analysis with adaptive exemplar selection. The framework comprises three components: (i) uncertainty exemplar selection; (ii) structure-aware entity and relation extraction; and (iii) final answer inference with LLMs. Figure 2 provides an overview of the workflow and a complete summary of all symbols is provided in Appendix D.

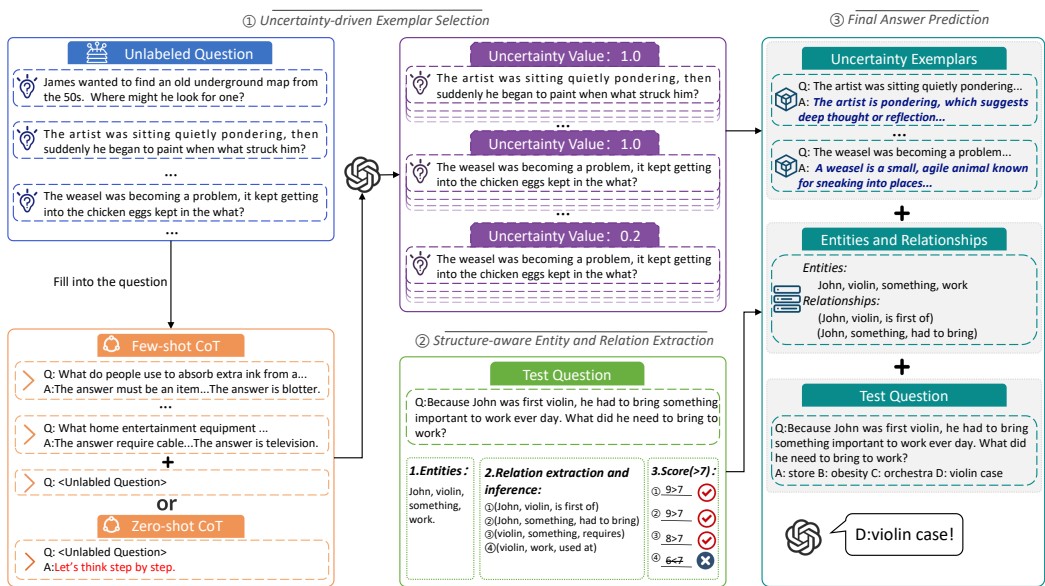

Figure 2: Overview of the SDA-CoT framework. (1) **Uncertainty-driven Exemplar Selection:** Using either few-shot or zero-shot CoT prompting, the unlabeled questions in the $\mathcal{D}_{\text{train}}$ are completed by leveraging the reasoning capabilities of the LLM to generate $m$ candidate answers for each question. The $k$ most uncertain questions are then selected into $E$ according to the uncertainty. (2) **Structure-aware Entity and Relation Extraction:** For each test input in $\mathcal{D}_{\text{test}}$, extract entities and infer both explicit and implicit relations among them. Use the LLM to analyze these relations and identify the valid ones. (3) **Final Answer Prediction**: High-quality annotated exemplars, extracted entities, valid relations, and test questions are fed into LLM for reasoning, which leads to the final answer.

## 3.1 UNCERTAINTY EXEMPLAR SELECTION

This step selects a small subset of highly uncertain questions from a large dataset to form the exemplar set $E$. Uncertainty in LLMs includes epistemic uncertainty (stemming from limited model knowledge) and aleatoric uncertainty (arising from data noise). Samples with high uncertainty often occur near decision boundaries or in out-of-distribution regions; labeling such samples helps reduce epistemic gaps and improves performance.

We first forward the LLM $m$ times to obtain $m$ answers for each question, and finally acquire the top-$k$ questions with the highest uncertainty. For stability, few-shot CoT prompting is applied using annotated examples from the training set $D_{\text{train}}$ (Wei et al., 2022a). Importantly, our framework does not rely on few-shot prompting: uncertainty is defined by the model's confidence distribution over outputs, which is independent of input format (Hüllermeier & Waegeman, 2021). CoT serves as a prefix intervention (i.e., $do(prefix = \text{CoT})$). As long as the core variables $X_{\text{core}}$ remain unchanged, the variance of the output distribution is unaffected (Zhang et al., 2025), making zero-shot prompting equally applicable.

Sample selection follows BAL, which chooses informative samples by maximizing expected information gain (Houlsby et al., 2011a):

$$x_i^* = \arg\max_{\mathbf{x}\in\mathcal{U}} I(\theta; y|\mathbf{x}, \mathcal{D}_{\text{train}}) = \arg\max_{\mathbf{x}\in\mathcal{U}} \mathbb{E} \tag{1}$$

where $\mathcal{U}$ is the unlabeled pool. To approximate the posterior $p(\theta \mid \mathcal{D})$ for LLMs, we perform $k$ stochastic forward passes with decoding randomness (e.g., top-$p$ sampling, temperature $\tau$). This yields outputs $\{y_1, y_2, \ldots, y_k\}$, which are used to compute three complementary metrics of uncertainty: disagreement, entropy, and variance.

**Disagreement.** Disagreement quantifies epistemic uncertainty by measuring variability among predictions under different posterior samples. Exact integration is intractable for LLMs, so we approximate $p(y \mid x_i, \mathcal{D}_{\text{train}})$ via $k$ stochastic forward passes. Given predictions $\{y_1, \ldots, y_k\}$, the score is:

$$\mathcal{U}_{\text{dis}}(x_i) = 1 - \frac{1}{k(k-1)} \sum_{i \neq j} \phi(y_i, y_j),$$ (2)

where $\phi$ is a task-specific semantic similarity function. This formulation captures variability across structured and unstructured outputs, consistent with epistemic exploration in BAL.

**Entropy.** Entropy measures the dispersion of the empirical output distribution. Given predictions $\{y_1, \ldots, y_k\}$ with empirical distribution $\hat{p}(y)$:

$$\mathcal{U}_{\text{ent}}(x_i) = -\sum_{y \in \mathcal{V}} \hat{p}(y) \log \hat{p}(y).$$ (3)

Low entropy reflects high confidence (predictions concentrate on one outcome), while high entropy signals diverse reasoning paths and ambiguous inputs, making such cases valuable for exemplar selection.

**Variance.** Variance evaluates stability in model confidence across sampled predictions. Each output $y_i$ is paired with a score $s_i$ (e.g., log probability or normalized likelihood). The variance is

$$\mathcal{U}_{\text{var}}(x_i) = \frac{1}{k-1} \sum_{i=1}^{k} (s_i - \bar{s})^2, \quad \bar{s} = \frac{1}{k} \sum_{i=1}^{k} s_i.$$ (4)

High variance indicates inconsistent confidence across outputs, exposing epistemic uncertainty and highlighting samples where calibration is most needed.

### 3.1.1 COMPOSITE SELECTION AND ANNOTATION

The three metrics are integrated into a weighted acquisition function:

$$\alpha(x_i) = \lambda_1 \mathcal{U}_{\text{dis}}(x_i) + \lambda_2 \mathcal{U}_{\text{ent}}(x_i) + \lambda_3 \mathcal{U}_{\text{var}}(x_i).$$ (5)

Here $\lambda_1$, $\lambda_2$, and $\lambda_3$ balance the importance of each signal. Samples with the highest $\alpha(x_i)$ are selected, manually annotated with reasoning chains, and added to the exemplar set. This human-in-the-loop process grounds exemplars in reliable logic and improves calibration, ultimately yielding more robust reasoning trajectories.

Table 1: Entity types.

| Type | Example sentences |
|------|-------------------|
| PEOPLE | **Turing** is a giant of computer science |
| ORGANIZATION | The **IPCC** warned about the cyclone. |
| LOCATION | The **Mt.Santias** loop is in Sunshine Canyon. |
| Geo-Political Entity | **Palo Alto** is raising the fees for parking. |
| Facility | Consider the **Golden Gate Bridge**. |
| Vechicle | It was a classic **Ford Falcon**. |

### 3.2 STRUCTURE-AWARE ENTITY AND RELATION EXTRACTION

**Entity extraction.** Building on the extraction capabilities of LLMs (Huang et al., 2025), each test question is supplied to the model with high-quality prompt instructions and the set of potential entity types in Table 1. For a given input $q_j$, the model lists entities $e_j$ contained in the question and assigns a predicted type to each $e_j$. Confidence is assessed with self-consistency (SC): for a candidate entity within $q_j$, the model produces $n$ independent extractions; the entity is retained when the vote count exceeds $\lceil n/2 \rceil$. This procedure filters out noisy or irrelevant items and produces the entity set $S$.

**Relation extraction and inference.** In zero-shot prompting, the objective is to map relations among entities. Explicit relations are directly stated in the text. For input $q_j$, the model extracts triples $(e_q, e_r, rel)$ and verifies them with SC, yielding the explicit set:

$$\mathcal{R}_e = \bigcup_{q,r \in \mathcal{S}} \{(e_q, e_r, rel)\} \tag{6}$$

Because $\mathcal{R}_e$ contains relations directly stated in the input, LLMs can read these pairs and then use them to support inference of pairs not written in the text.

After collecting explicit relations, the next step is to infer implicit pairs so that the model can recover the relational structure of the sentence. An implicit pair is not stated verbatim but is supported by $S$ and $\mathcal{R}e$. For example: *"Because John was first violin, he had to bring something important to work every day. What did he need to bring to work?"* The link between John and violin is explicit and can be extracted, whereas the link between violin and something is unstated and must be inferred using $S$ and $\mathcal{R}e$. Since multi-step reasoning is required, additional information is provided: the query $q_j$ together with $\mathcal{R}e$ is given to the model, which returns $l$ implicit relations ranked by relevance. Let an intermediate step be $T_a = (e_a, e_{a+1}, rel_a)$; such steps form a chain $T_1, T_2, \ldots, T_{n-1}$ that yields an interpretable triple $T_{1 \to n} = (e_1, e_n, rel_x)$, where $n$ is the chain length. The inferred set is:

$$\mathcal{R}'_i = \bigcup_{q,r \in \mathcal{S}} \{(e_q, e_r, rel_x)\} \tag{7}$$

where $rel_x$ denotes relations produced during the reasoning step.

**Relation validation.** To ensure reliability, we adopt Self-Correction (Yan et al., 2025), treating the LLM as a scoring function. A reliability threshold $sco_{th}$ is fixed. The model receives the original text together with all candidate relations and, for each triple, returns a score $sco(i, j, k)$ that reflects its plausibility. Triples with higher scores are treated as more likely to be correct. Relations that score below the threshold are removed because they would misguide downstream reasoning, whereas those at or above the threshold are kept as reliable instances. The resulting set of implicit relations is:

$$\mathcal{R}_i = \bigcup_{q,r \in \mathcal{S}, sco(i,j,k) \geq sco_{th}} \{(e_q, e_r, rel_x)\} \tag{8}$$

which records all retained pairs $(e_q, e_r)$ together with the inferred label $rel_x$.

### 3.3 INFERENCE

The final stage concerns answer inference, where previously derived information is combined and used to make predictions. Three sources are used: (i) high-quality exemplars $E$; (ii) the entity set $\mathcal{S}$; and (iii) the validated relation set $\mathcal{R}_{\text{valid}} = \mathcal{R}_e \cup \mathcal{R}_i$. Together, these components provide a structured context that guides the LLM during reasoning. Formally, for a test query $q_j$, the input prompt to the LLM $\mathcal{M}$ is:

$$\text{Prompt}(q_j) = \{E, \mathcal{S}, \mathcal{R}_{\text{valid}}, q_j\}, \tag{9}$$

where exemplars $E$ supply reasoning patterns, entities $\mathcal{S}$ anchor the problem context, and relations $\mathcal{R}_{\text{valid}}$ give the links required for multi-hop inference. With this context, the model conditions its reasoning on both the surface text and the structured semantic information.

To further stabilize predictions, we employ SC. The LLM is queried $n$ times with stochastic decoding, producing reasoning chains $\{y_1, \ldots, y_n\}$. The final answer is obtained by majority vote:

$$y^* = \underset{y \in \{y_1, \ldots, y_n\}}{\arg\max} \ \text{Count}(y), \tag{10}$$

where $\text{Count}(y)$ denotes the frequency of prediction $y$. This mechanism suppresses spurious reasoning and amplifies the most reliable inference trajectory.

## 4 EXPERIMENTS

### 4.1 EXPERIMENTAL SETUP

**Datasets and models.** To evaluate the effectiveness and generalizability of SDA-CoT, we conduct experiments spanning three categories of reasoning: commonsense reasoning, logical reasoning, and

mathematical problem solving. Specifically: (i) for commonsense reasoning, we use StrategyQA (StrQA) (Geva et al., 2021) and CommonsenseQA (CSQA) (Talmor et al., 2019), which test the model's ability to leverage implicit knowledge beyond surface text; (ii) for logical reasoning, we adopt LogiQA (Liu et al., 2020), HotpotQA (HPQA) (Yang et al., 2018), and 2WikiMultiHopQA (2WikiQA) (Ho et al., 2020), which evaluate multi-hop and structurally complex reasoning; (iii) for math word problems, we use GSM8K (Cobbe et al., 2021), a standard benchmark for assessing chain-of-thought reasoning in mathematical contexts. Table 5 summarizes these datasets.

We evaluate SDA-CoT using two representative LLMs: LLaMA2-13B(Touvron et al., 2023) and DeepSeek-R1 (DeepSeek-AI et al., 2025), to evaluate the proposed SDA-CoT framework. Specifically, LLaMA2-13B is a transformer-based language model, and it has been widely adopted as a strong baseline for natural language understanding and reasoning tasks. Deepseek-R1 is a reasoning-centered LLM that incorporates mechanisms encouraging deep thinking, making it particularly well-suited for complex, multi-hop, and mathematical reasoning tasks.

**Baselines.** To ensure a comprehensive evaluation, we compare against several representative baselines. **Vanilla LM**, which relies solely on in-context learning without reasoning traces; **Chain-of-Thought (CoT)** (Wei et al., 2022b) improves reasoning by generating explicit intermediate steps, while **CoT-SC** (Wang et al., 2023b) enhances its robustness via majority voting across multiple reasoning paths; We also consider more advanced variants such as **Auto-CoT** (Zhang et al., 2023a), which automatically selects exemplars, and **Complex-CoT** (Fu et al., 2023a), which adapts reasoning to task difficulty; **Plan-and-Solve (PS)** (Wang et al., 2023a) further structures reasoning into a planning phase followed by detailed problem solving; **CDW-CoT** (Fang et al., 2025), which dynamically constructs prompts through clustering and distance-weighted selection.

## 4.2 EXPERIMENTAL ANALYSIS

### 4.2.1 MAIN RESULT

Table 2: Main results (%). The best score are highlighted in **bold**, and the second-best one is underlined. CoT-SC@5 represents retrieving five CoT reasoning chains when applying SC principles.

| Model | Methods | CSQA | StrQA | LogiQA | HPQA | 2WikiQA | GSM8K |
|-------|---------|------|-------|--------|------|---------|-------|
| LLaMA2-13B | Vanilla LM | 57.2 | 58.3 | 24.5 | 34.2 | 28.2 | 17.8 |
| | CoT | 55.1 | 64.2 | 30.2 | 37.1 | 32.4 | 18.9 |
| | CoT-SC@5 | 57.2 | 66.8 | 32.4 | 36.8 | 34.6 | 21.2 |
| | Auto-CoT | 56.8 | 66.5 | 31.9 | 37.5 | 35.2 | _25.1_ |
| | Complex-CoT | 54.8 | 65.2 | 32.1 | 37.1 | 35.1 | 23.8 |
| | PS | 56.8 | 66.2 | 31.6 | 36.9 | 34.2 | 22.4 |
| | CDW-CoT | _61.4_ | _70.1_ | _33.4_ | _40.1_ | _37.6_ | 24.1 |
| | SDA-CoT(ours) | **67.5** | **78.5** | **41.6** | **45.3** | **44.9** | **37.5** |
| DeepSeek-R1 | Vanilla LM | 63.6 | 69.4 | 26.8 | 41.7 | 36.6 | 49.0 |
| | CoT | 61.4 | 74.5 | 35.0 | 44.4 | 42.1 | 66.9 |
| | CoT-SC@5 | 63.3 | 75.6 | 36.8 | 44.8 | 44.8 | 71.3 |
| | Auto-CoT | 62.8 | 74.9 | 37.2 | 45.1 | 45.1 | 73.6 |
| | Complex-CoT | 62.4 | 73.5 | 37.2 | 44.5 | 44.5 | _76.5_ |
| | PS | 63.9 | 74.8 | 36.4 | 45.2 | 45.2 | 72.6 |
| | CDW-CoT | _69.4_ | _80.1_ | _43.0_ | _49.0_ | _46.2_ | 74.2 |
| | SDA-CoT(ours) | **76.2** | **86.7** | **50.3** | **53.0** | **52.5** | **80.0** |

Table 2 summarizes the results across six benchmarks, SDA-CoT consistently outperforms all baselines, including the strong adaptive method CDW-CoT. On average, SDA-CoT improves performance by 9% with LLaMA2-13B and by 8% with DeepSeek-R1. The largest improvement occurs on GSM8K, where SDA-CoT improves accuracy by over 12% on LLaMA2-13B.

**Mechanistic perspective.** SDA-CoT improves reasoning through two complementary mechanisms. The uncertainty exemplar selection prioritizes informative cases where the framework exhibits high predictive variance. This exposes the model to diverse, challenging reasoning patterns and prevents overfitting to trivial examples. In parallel, entity and relation extraction improves the framework's ability to capture relational dependencies within a question. By explicitly extracting both explicit and implicit entities and relations, SDA-CoT establishes a structured reasoning context that aligns with CoT generation. This is especially beneficial for multi-hop reasoning and arithmetic decomposition, where reasoning must go beyond surface level text.

**Empirical insights.** Conventional CoT and its variants improve reasoning quality compared to vanilla LM, yet their performance is limited, especially on HPQA and 2WikiQA. This suggests that simply increasing reasoning paths is insufficient to capture the dependencies required for complex reasoning. SDA-CoT addresses this limitation by incorporating entity and relation structures, enabling the framework to dynamically adapt reasoning steps to the problem. Notably, this advantage is markedly enhanced in DeepSeek-R1, which has been specifically optimized for reasoning.

**Impact of LLMs.** The two LLMs highlight complementary aspects of SDA-CoT. LLaMA2-13B, as an open source model, demonstrates the fundamental effectiveness of our approach even without specialized reasoning mechanisms. DeepSeek-R1, designed for reasoning tasks, shows that SDA-CoT can further enhance models already optimized for deep reasoning.

### 4.2.2 ABLATION STUDY

Table 3: Ablation study.

| Model | Methods | CSQA | StrQA | LogiQA | HPQA | 2WikiQA | GSM8K |
|-------|---------|------|-------|--------|------|---------|-------|
| | w/o ES | 60.2 | 71.5 | 34.4 | 40.3 | 39.2 | 28.1 |
| | Only EE | 63.7 | 71.1 | 31.8 | 41.3 | 39.8 | 35.1 |
| LLaMA2-13B | w/o ERI | 64.4 | 74.5 | 33.2 | 42.3 | 40.4 | 36.9 |
| | w/o ERE | 64.9 | 73.7 | 33.7 | 42.6 | 40.7 | 37.8 |
| | SDA-CoT | 67.5 | 78.5 | 41.6 | 45.3 | 45.0 | 37.5 |
| | w/o ES | 70.8 | 80.2 | 43.7 | 50.2 | 47.3 | 73.2 |
| | Only EE | 71.5 | 81.1 | 41.4 | 48.6 | 48.0 | 79.3 |
| DeepSeek-R1 | w/o ERI | 74.5 | 83.9 | 46.2 | 50.6 | 50.9 | 80.4 |
| | w/o ERE | 73.9 | 84.5 | 46.9 | 50.7 | 50.2 | 79.6 |
| | SDA-CoT | 76.2 | 86.7 | 50.3 | 53.0 | 52.5 | 80.0 |

We perform an ablation study to quantify the contribution of the principal components in SDA-CoT. Table 3 reports results under four reduced settings: (i) **w/o ES** — removing uncertainty exemplar selection; (ii) **Only EE** — retaining only entity extraction, without relation modeling; (iii) **w/o ERI** — removing implicit relation inference; and (iv) **w/o ERE** — removing explicit relation extraction.

**Exemplar selection.** Removing the ES causes the largest performance decline, with accuracies dropping by 7%–9% across both LLMs. This confirms that exemplar selection is crucial, as it ensures exposure to diverse and high uncertainty exemplar. Declines are particularly sharp on GSM8K, underscoring its importance for math word problems.

**Structural reasoning.** EE, ERI, and ERE each provide complementary gains. When structural reasoning is partially disabled, the model loses between 3%–5% on average. EE provides a strong foundation for tasks requiring numerical alignment, as explicit extraction of entities and quantities can be helpful for math problems. Implicit relation inference is especially critical for logical and multi-hop datasets, where unstated but derivable relations form the LLM of multi-step reasoning. Explicit relation extraction provides stable anchors that constrain reasoning paths, reducing error spreading and improving the accuracy of CoT.

**Interaction effects.** The full SDA-CoT consistently outperforms some variants by 6%–10%, demonstrating complementary between exemplar selection and structural reasoning. High uncer-

Table 4: Accuracy (%) of SDA-CoT on sentences with few entities and relations (entities $\leq 2$, relations $\leq 2$) across different reasoning categories. Results are reported with and without Uncertainty exemplar selection module. The experiment is conducted based on LLaMA2-13B.

| Setting | CSQA | StrQA | LogiQA | HPQA | 2WikiQA | GSM8K |
|---|---|---|---|---|---|---|
| *Entity extraction accuracy* | | | | | | |
| SDA-CoT (with ES) | 65.2 | 76.1 | 39.8 | 43.2 | 42.1 | 36.0 |
| SDA-CoT (w/o ES) | 63.9 | 74.8 | 38.5 | 41.9 | 40.7 | 34.7 |
| *Explicit relation accuracy* | | | | | | |
| SDA-CoT (with ES) | 61.7 | 72.8 | 37.1 | 40.5 | 39.2 | 32.8 |
| SDA-CoT (w/o ES) | 60.1 | 71.2 | 35.6 | 39.0 | 37.8 | 31.4 |
| *Implicit relation accuracy* | | | | | | |
| SDA-CoT (with ES) | 55.9 | 67.5 | 33.4 | 36.1 | 35.0 | 29.7 |
| SDA-CoT (w/o ES) | 54.2 | 65.9 | 31.8 | 34.7 | 33.5 | 28.2 |
| SDA-CoT (with ES) | 58.0 | 69.8 | 35.0 | 38.0 | 36.8 | 30.9 |

tainty exemplars supply rich clues, while entity and relation modeling provide structure, together forming a feedback loop that enhances reliability and interpretability.

### 4.2.3 ANALYSIS OF SENTENCES WITH FEW ENTITIES AND RELATIONS

During the experiments, we observed that SDA-CoT faces limitations when handling simple sentences with "few entities and relations": the chains tend to be short, structural evidence is limited, and relation extraction is often incomplete. As shown in Table 4, when sentences contain only a small number of entities and relations, the accuracy of implicit relation inference is constrained by the chain effect of entity count and explicit relation count. Although implicit relation accuracy decreases under these conditions, SDA-CoT still achieves state-of-the-art performance compared with other CoT methods. This advantage arises because entity and relation extraction transforms sparse semantic cues into structured schemas, thereby constraining reasoning paths under low-connectivity conditions. At the same time, uncertainty exemplar selection supplements the model with highly uncertain samples, allowing it to access diverse reasoning examples even when evidence is insufficient.

On the LogiQA dataset, entity recognition accuracy increased from 38.5% to 39.8%, and implicit relation accuracy rose from 31.8% to 33.4%. On GSM8K, entity extraction and implicit relation accuracy improved by 1.3% and 1.5%, respectively. Although the gains are modest, they reduce fluctuations in performance on low-density inputs. These results indicate that even simple sentences may still contain deep level entity relations.

## 5 CONCLUSION

This paper presents SDA-CoT, a structure-driven and dynamically adaptive framework for CoT. It combines uncertainty-aware exemplar selection with structure-based entity and relation extraction to reduce the sensitivity of CoT to exemplar phrasing and to improve performance on tasks with long texts or many complex entity relations. Experiments on multiple benchmarks show that SDA-CoT improves reasoning accuracy and generalization and outperforms prior CoT methods. The results indicate that structural signals guide LLM reasoning and can reduce the fragility of the exemplar design. The method can add computational overhead during structure extraction. Future work will study lighter structural representations. These directions may further raise the reliability and interpretability of LLM reasoning.

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

# A  DATASETS

Table 5 presents detailed information on the datasets used in the experiments. Because the CSQA, HPQA, 2WikiQA, and GSM8K datasets are extremely large (e.g., 192,606 entries in 2WikiQA), we randomly sampled from them to balance experimental cost and evaluation effectiveness.

Table 5: Overview of the datasets.

| Dataset | Domain / Task | Size |
|---------|---------------|------|
| StrQA | Commonsense reasoning | 2061 |
| CSQA | Commonsense reasoning | 5761 |
| LogiQA | Logical reasoning | 7424 |
| HPQA | Logical reasoning | 5008 |
| 2WikiQA | Logical reasoning | 1900 |
| GSM8K | Math word problems | 4644 |

# B  IMPLEMENTATION

## B.1  HARDWARE DEVICES AND PARAMETER CONFIGURATION

Table 6: Experimental environment and parameter configuration.

| Category | Description |
|----------|-------------|
| Programming Language | Python 3.10 |
| Operating System | Windows 10 |
| Processor | 12th Gen Intel(R) Core(TM) i9-12900H |
| GPU Configuration | $2 \times$ NVIDIA A100 GPUs |

The experimental setup was designed to ensure both reproducibility and computational efficiency. As shown in Table 6, all experiments were implemented in Python 3.10 on Windows 10, using a 12th Gen Intel Core i9-12900H processor and two NVIDIA A100 GPUs to support large-scale model operations.

The DeepSeek-R1 model was accessed through the SiliconFlow API[1] , and the open-source LLaMA2-13B model was obtained locally for experimentation. Within the proposed framework, the temperature was set to 0.1 for entity extraction and explicit extraction, 0.4 for implicit extraction, and 0.7 during the final inference stage. This configuration follows the principle that lower temperatures yield more definitive outputs, while higher temperatures encourage greater diversity in reasoning. For exemplar selection, we adopted the method of Wei et al. (2022b). The number of exemplars was fixed at 6 for StrQA, 7 for CSQA, and 8 for LogiQA, HPQA, 2WikiQA, and GSM8K. SC was applied in two stages: relation extraction and inference. Each implicit relation pair was evaluated five times, and during the reasoning stage, each question was processed ten times. To validate implicit relation pairs, we set the threshold $sco_{th}$ to 7, retaining only high quality pairs. When no pair reached this value, the three pairs with the highest scores were selected instead.

## B.2  EVALUATION METRIC

We use accuracy and exact match as the evaluation metric for different datasets. Specifically, for multiple-choice datasets such as StrQA, CSQA, and LogiQA, accuracy is computed by checking whether the selected option matches the ground-truth answer. For problems like GSM8K, where the output is a number, we use regular expressions for exact match judgment of the answers. For open-ended datasets like 2WikiMQA that do not contain question options, predictions are compared against answer alternatives using the exact match criterion. The same processing approach is adopted for different methods across these datasets.

---

[1] https://siliconflow.cn/

## B.3 PROMPT TEMPLATE

The following is the prompt template used by SDA-CoT to structure the CoT. All prompts are used in a zero-shot setting and operate solely on the initial questions.

---

**Entities extraction**

---

Extract all named entities from the given sentence, possible entities may include: [individuals, organizations, locations, ..., percentages].

**Sentence**: [ $q_j$ ]

**Entities**: [ ]

---

**Explicit relation extraction**

---

Given a sentence and its entities, extract all explicitly stated relations as triples in the form: (Entity1, Entity2, Rel). Output only triples, without explanations or extra text.

**Sentence**: [ $q_j$ ]

**Entities**: [ $E_j$ ]
**Relations**: [ ]

---

**Implicit relation extraction**

---

Given a sentence, its entities, and all explicit relations, infer possible implicit relations between entities. For each entity pair, output implicit relations as triples in the form: (Entity1, Entity2, Relation). Output only triples, without explanations or extra text.

**Sentence**: [ $q_j$ ]

**Entities**: [ $E_j$ ]

**Explicit relations**: [ $\mathcal{R}_e$ ]

**Implicit relations**: [ ]

---

**Relation validation**

---

Given a sentence and all implicit relations, assign a confidence score from 0 to 10, where higher values indicate greater likelihood of correctness. Output each as a quadruple: (Entity1, Entity2, Relation, Score). Output only results, without explanations or extra text.

**Sentence**: [ $q_j$ ]

**Entities**: [ $E_j$ ]

**Implicit relations**: [ $\mathcal{R}_i$ ]

**Implicit relation scores**: [ ]

---

> **Inference**
>
> ---
>
> Given a sentence, its entities, and all relations, answer the question based only on this information.
>
> **Sentence**: [ $q_j$ ]
>
> **Entities**: [ $E_j$ ]
>
> **Explicit relations**: [ $\mathcal{R}_e$ ]
>
> **Implicit relations**: [ $\mathcal{R}_i$ ]
>
> **Answer**:

## C  DISCUSSIONS

### C.1  EFFECTS OF POOL SIZE.

Table 7: Performance of DeepSeek-R1 with varying numbers of sampled generations.

| Number | StrQA | HPQA | 2WikiMHQA | GSM8K |
|--------|-------|------|-----------|-------|
| 1 | 78.1 | 43.9 | 48.6 | 70.4 |
| 5 | 82.8 | 50.4 | 50.3 | 77.2 |
| 10 | 86.7 | 53.0 | 52.5 | 80.0 |
| 15 | 89.9 | 53.5 | 53.1 | 81.5 |

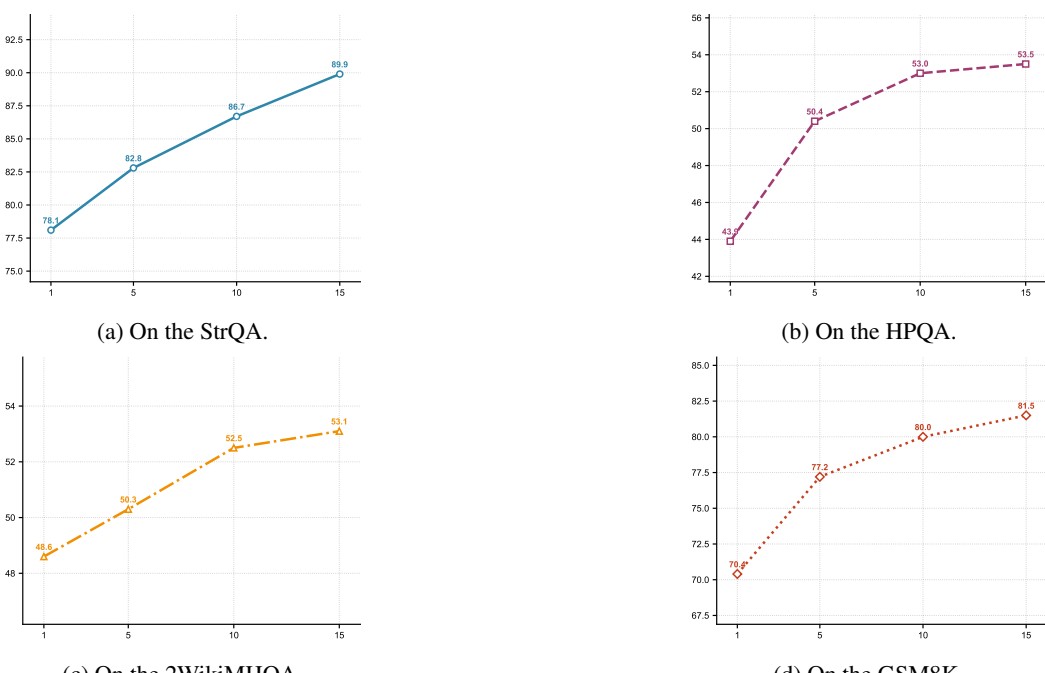

(a) On the StrQA.

(b) On the HPQA.

(c) On the 2WikiMHQA.

(d) On the GSM8K.

Figure 3: Performance of DeepSeek-R1 with different numbers of sampled generations.

We investigate how the number of sampled model generations $m$ used for SC influences downstream reasoning accuracy. Results in Table 7 and Figure 3 show that increasing $m$ generally improves performance, as larger pools yield more reliable uncertainty estimates and higher-quality exemplar selection. Better exemplars enhance few-shot prompts, compounding the benefits of SC itself.

In our pipeline, we use multiple generations to estimate per-instance uncertainty (disagreement/entropy/variance). Larger $m$ yields more reliable uncertainty estimates, improving the selection of high-value exemplars for annotation. Better exemplars produce higher-quality few-shot prompts during inference, amplifying downstream benefits—this effect compounds the direct gains obtained from SC alone. The trade-off is computational cost versus robustness. Increasing $m$ linearly raises inference overhead, while providing sublinear accuracy returns after a certain point. Based on the presented results, we recommend $m = 10$ as a default: it captures most of the attainable improvement with reasonable cost.

Our experiments reveal that arithmetic reasoning (GSM8K) exhibit larger sensitivity to $m$, reflecting the need to explore diverse numerical decompositions and implicit inference chains. Multi-hop reading comprehension tasks (HotpotQA, 2WikiMHQA) tend to saturate earlier, suggesting that a modest ensemble of chains often suffices to capture the necessary evidence aggregation behavior.

## C.2 Effects of Self-Consistency.

To disentangle the role of SC, we evaluate SDA-CoT under two settings: (i) the full SDA-CoT framework with SC applied during inference, and (ii) a variant *w/o SC*, where the final-stage SC mechanism is removed while all previous stages remain intact. The results are reported in Figure 4.

These results highlight two distinct roles of SC within SDA-CoT. Specifically, in exemplar selection stage, SC operates through the generation of multiple model outputs (denoted as $m$), which guides the choice of informative exemplars for subsequent stages. In contrast, during inference stage, SC aggregates multiple answer samples ($n$) to produce a more stable and reliable prediction. Although both implementations of SC share a common reliance on sampling diverse CoT trajectories, their contributions to the overall reasoning pipeline are inherently complementary: exemplar selection stage leverages SC to refine exemplar quality, whereas inference uses SC to stabilize the final output.

The observed performance improvements attributed to inference stage further validate the necessity of this latter application of SC: even after integrating exemplar-driven prompting and structure-aware reasoning—two core components designed to refine the reasoning process—residual variability in the model's final predictions remains non-negligible, and this variability can be effectively mitigated through the deployment of SC.

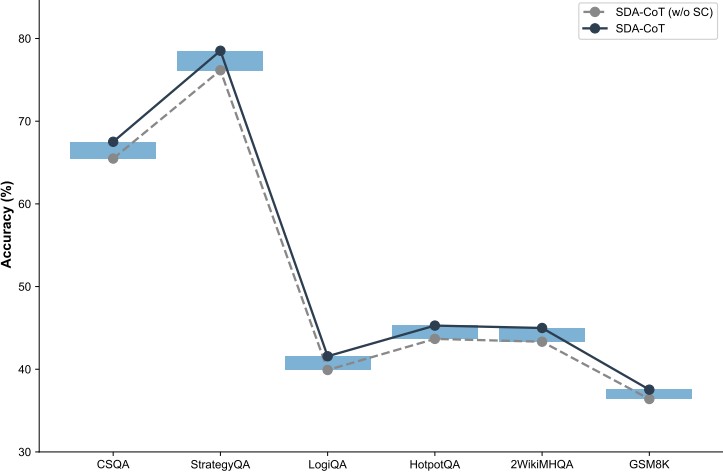

Figure 4: Effects of self-consistency

## C.3 Case Study

To provide a clearer understanding of how SDA-CoT improves reasoning compared with standard CoT, we present case studies on two representative tasks: HotpotQA and GSM8K.

Table 8: Illustrative HotpotQA example comparing CoT and SDA-CoT reasoning chains.

| Method | Reasoning Chain (excerpt) |
|---|---|
| CoT | "Document A mentions $X$. Document B mentions $Y$. Therefore, the answer is $Z$." (Incorrect: failed to connect $X$ and $Y$) |
| SDA-CoT | "Extract entity $X$ from Document A. Entity $Y$ from Document B is related via [explicit relation]. Implicitly, $X$ is connected to $Y$ through [reasoned link]. Therefore, the answer is $Z$." (Correct) |

Table 9: Illustrative GSM8K example comparing CoT and SDA-CoT reasoning on a math word problem. Exemplar selection helps the model avoid spurious reasoning even under single-sample inference.

| Method | Reasoning Chain (excerpt) |
|---|---|
| Baseline CoT | "John had 25 apples, he gave away 7, then he had $25 + 7 = 32$." (Incorrect: arithmetic slip due to misapplied operation) |
| SDA-CoT | "Exemplar-guided prompt highlights subtraction in similar problems. Entity extraction: John (25 apples), gave away (7). Correct relation: subtraction. Step: $25 - 7 = 18$. Therefore, John has 18 apples left." (Correct: exemplar selection + structural reasoning) |

**Multi-hop reasoning (HotpotQA).** Table 8 shows an example that requires aggregating evidence across two documents. CoT often retrieves the correct entities but fails to link them coherently, leading to an incorrect answer. In contrast, SDA-CoT first extracts entities and explicit relations, then infers implicit relations to connect intermediate facts. This module yields a consistent chain, reinforced by SC in the final stage. The result is a correct prediction with improved interpretability. Notably, entity and relation structures prevent the model from drifting into irrelevant reasoning paths, which we observed in more than 30% of CoT outputs.

**Arithmetic reasoning (GSM8K).** Math word problems in GSM8K are highly sensitive to reasoning errors, since even small slips can yield incorrect answers while the reasoning chain appears plausible. Table 9 illustrates a typical error: the CoT incorrectly applies addition instead of subtraction. SDA-CoT avoids this mistake through two mechanisms:

- Uncertainty exemplar selection ensures that high uncertainty math word problems are included in the exemplar pool. These exemplars expose the model to diverse solution patterns and guide it away from spurious reasoning in single question inference.

- entity extraction identifies numerical quantities, while relation modeling ensures the correct arithmetic operation is applied.

At the final stage, SC further stabilizes predictions by aggregating reasoning chains. Even without aggregation, prompts substantially reduce single question errors. This demonstrates that exemplar selection and structural reasoning jointly improve both robustness and accuracy in math word problems.

The core advantage of SDA-CoT is that structure-aware reasoning constrains the reasoning path, reducing spurious or inconsistent outputs. Furthermore, uncertainty-based exemplar selection provides informative demonstrations that improve robustness. These effects are reinforced by SC, which filters noise during inference. Together, these mechanisms yield not only higher accuracy but also more stable and interpretable reasoning processes.

# D NOTATION

For clarity, we provide a detailed summary of all notation used in the Method section. Table 10 serves as a complete reference.

Table 10: Summary of notation used in the proposed framework.

| Symbol | Description |
|---|---|
| $\mathcal{D}_{\text{train}}$ | Labeled training dataset |
| $\mathcal{D}_{\text{test}}$ | Test dataset |
| $\mathcal{U}$ | Unlabeled pool of questions |
| $E$ | Exemplar set selected from $\mathcal{D}_{\text{train}}$ |
| $m$ | Number of candidate answers per question |
| $k$ | Number of top uncertain questions retained |
| $s_i$ | Confidence score associated with prediction $y_i$ |
| $q_j$ | A test query |
| $e_j$ | Entity extracted from question $q_j$ |
| $\mathcal{S}$ | Extracted entity set |
| $\mathcal{R}_e$ | Set of explicit relations extracted from the text |
| $\mathcal{R}_i'$ | Candidate set of inferred implicit relations |
| $\mathcal{R}_i$ | Validated set of implicit relations |
| $sco_{th}$ | Threshold for filtering reliable relations |
| $\mathcal{R}_{\text{valid}}$ | Final validated relation set, $\mathcal{R}_e \cup \mathcal{R}_i$ |
| $\mathcal{M}$ | The large language model (LLM) |

# E ALGORITHM OF UNCERTAINTY EXEMPLAR SELECTION

Algorithm 1 provides the description of the proposed uncertainty-driven exemplar selection procedure. The goal is to construct a compact exemplar set $E$ from the training set $\mathcal{D}_{\text{train}}$ by prioritizing samples with high epistemic and aleatoric uncertainty. The algorithm integrates disagreement, entropy, and variance into a weighted acquisition function, followed by human annotation of the selected exemplars. Inputs include the training data, a large language model $\mathcal{M}$, decoding parameters, and the number $K$ of exemplars to select. The output is an annotated exemplar set $E$ used for few-shot reasoning and calibration.

---

**Algorithm 1** Uncertainty-Driven Exemplar Selection

---

**Require:** Training set $\mathcal{D}_{\text{train}}$, number of generations $k$, top-$K$ selection size, LLM model $\mathcal{M}$, CoT strategy (`few-shot` or `zero-shot`), uncertainty weights $\lambda_1, \lambda_2, \lambda_3$
**Ensure:** Annotated exemplar set $E$
1: Initialize $E \leftarrow \emptyset$
2: **for** each question $x_i \in \mathcal{D}_{\text{train}}$ **do**
3:     Apply CoT to form input prompt $q_i$ using `CoT strategy`
4:     Generate $k$ reasoning chains $\{y_1, y_2, \ldots, y_k\}$ from $\mathcal{M}(q_i)$ via stochastic decoding
5:     Compute pairwise semantic similarity $\phi(y_j, y_l)$ for all $j \neq l$
6:     $\mathcal{U}_{\text{dis}}(x_i) \leftarrow 1 - \frac{1}{k(k-1)} \sum_{j \neq l} \phi(y_j, y_l)$
7:     Estimate output frequency $\hat{p}(y)$ and compute entropy:
8:     $\mathcal{U}_{\text{ent}}(x_i) \leftarrow - \sum_{y \in \mathcal{V}} \hat{p}(y) \log \hat{p}(y)$
9:     Compute confidence scores $\{s_j\}$ for each $y_j$, then:
10:     $\bar{s} \leftarrow \frac{1}{k} \sum_{j=1}^{k} s_j$
11:     $\mathcal{U}_{\text{var}}(x_i) \leftarrow \frac{1}{k-1} \sum_{j=1}^{k} (s_j - \bar{s})^2$
12:     Compute combined uncertainty score:
13:     $\alpha(x_i) \leftarrow \lambda_1 \mathcal{U}_{\text{dis}}(x_i) + \lambda_2 \mathcal{U}_{\text{ent}}(x_i) + \lambda_3 \mathcal{U}_{\text{var}}(x_i)$
14: **end for**
15: Select top-$K$ samples $S = \{x_i \mid \text{highest } \alpha(x_i)\}$
16: **for** each $x_i \in S$ **do**
17:     Manually annotate reasoning chain and final answer $(\text{CoT}_i, a_i)$
18:     $E \leftarrow E \cup \{(x_i, \text{CoT}_i, a_i)\}$
19: **end for**
        **return** $E$

---

## F  THE USE OF LARGE LANGUAGE MODELS (LLMS)

We used a large language model only to refine the written paragraphs, making them more fluent and readable. No other aspects of the work used large language models beyond this text refinement.

## G  CODE

Our implementations for all experiments is available at `https://anonymous.4open.science/r/SDA-CoT-F6D6/`.

