# OpenReview forum: "SDA-CoT: Structure-driven Dynamic Active Chain-of-Thought"
_ICLR.cc/2026/Conference — ICLR 2026 Conference Withdrawn Submission_

### Official Review · Reviewer_yv9m · 2025-10-21

**Soundness:** 3
**Presentation:** 3
**Contribution:** 3
**Rating:** 8
**Confidence:** 4

**Summary:**

This paper introduces SDA-CoT, a framework designed to improve Chain-of-Thought (CoT) for Large Language Models (LLMs) reasoning. To address two limitations of many other CoT methods, reliance on fixed exemplars and neglect of structural cues in the text, SDA-CoT integrates three components: 1) a Bayesian Active Learning (BAL) module that dynamically selects high-uncertainty exemplars for annotation, using a composite of disagreement, entropy, and variance; 2) a structure extraction module that identifies entities and infers both explicit and implicit relations from the input question; and 3) an inference step that combines the selected exemplars and the extracted structure to generate a reasoning prompt. Experiments across six benchmarks (StrategyQA, CSQA, LogiQA, HotpotQA, 2WikiMultiHopQA, GSM8K) of two LLMs (LLaMA2-13B and DeepSeek-R1) demonstrate state-of-the-art performance, with an average accuracy improvement of 9% and 8% over strong baselines.

**Strengths:**

* Addresses core challenges of CoT methods in LLM reasoning
* Introduces the composite uncertainty metric
* Focuses on implicit relation inference
* The experimental evaluation is extensive and the performance gains are convincing
* Writing is clear and the structure is logical

**Weaknesses:**

* The computational cost is not quantified. It would be helpful to know the additional cost (especially inference time) compared with standard CoT methods.
* Human annotation of the selected exemplars certainly strengthens the performance, but the annotation cost is not analyzed.
* The analysis of sentences with few entities/relations (Section 4.2.3) is interesting, but it also demonstrates the inherent limitation of such structure-aware approaches.

**Questions:**

1. Could you please provide a more detailed analysis of the computational cost (e.g., total inference time) of the full SDA-CoT pipeline compared to a standard CoT baseline?
2. Have you explored or considered strategies to automate the human annotation, for instance, by using a much larger LLM (like GPT-4) as the "oracle" to generate the high-quality CoTs?
3. Could you please elaborate on which types of errors the entity and relation modules effectively prevent? (Table 3: Ablation study)

---

### Official Review · Reviewer_xu85 · 2025-10-29

**Soundness:** 2
**Presentation:** 2
**Contribution:** 2
**Rating:** 2
**Confidence:** 4

**Summary:**

The paper presents SDA-CoT, a method that improves large language models’ reasoning by combining uncertainty-based exemplar selection and structure-aware reasoning.
It uses Bayesian Active Learning to pick high-uncertainty questions for manual CoT annotation and extracts entities and relations from each test question to build structured prompts.
Experiments on six benchmarks with LLaMA2-13B and DeepSeek-R1 show consistent gains over several CoT baselines, about 8–9 points on average and 13 points on GSM8K.
The approach improves reasoning accuracy and stability but adds computational cost due to structure extraction.

**Strengths:**

1.The paper proposes a clear and well-motivated idea that combines uncertainty-based exemplar selection with structure-aware reasoning.
2.The method design is logical, using Bayesian Active Learning to select informative examples and entity–relation extraction to guide reasoning.
3.Experiments are broad and well executed on six datasets and two LLMs, with consistent improvements over baselines.
4.Writing and organization are clear, and figures help explain the method effectively.

**Weaknesses:**

1.The work has limited novelty. Similar ideas have appeared in CoK (2023) and ERA-CoT (2024), which also combine structured reasoning with CoT. The main addition here is using Bayesian Active Learning for exemplar selection.
2.Some implementation details are missing, such as the definition of the similarity function ϕ, confidence score computation, and λ weights.
3.The evaluation lacks statistical tests, ground-truth validation for entity/relations, and a cost-benefit analysis.
4.Minor presentation issues remain (typos, inconsistent dataset names).

**Questions:**

1. Have you compared SDA-CoT with other recent uncertainty-based or structure-aware CoT approaches?
For example, Active Prompting with Chain-of-Thought for Large Language Models (Diao et al., 2023) also uses uncertainty to select informative exemplars, and Chain-of-Knowledge Prompting (Wang et al., 2023) or ERA-CoT (Liu et al., 2024) integrate entity–relation or knowledge-triple structures into CoT reasoning.

---

### Official Review · Reviewer_ndcu · 2025-10-30

**Soundness:** 2
**Presentation:** 3
**Contribution:** 1
**Rating:** 2
**Confidence:** 4

**Summary:**

This paper introduces SDA-CoT, a framework that enhances LLM reasoning by combining structural information with uncertainty-based exemplar selection. According to the authors, unlike standard CoT methods that rely on fixed human-written exemplars, SDA-CoT dynamically selects the most informative examples using Bayesian Active Learning and builds entity–relation graphs to guide reasoning. The framework consists of three key modules: (1) uncertainty-driven exemplar selection, (2) structure-aware entity and relation extraction, and (3) CoT reasoning with structural context. Experiments on several reasoning benchmarks (e.g., StrategyQA, LogiQA, GSM8K) using LLaMA2-13B and DeepSeek-R1 show improvements.

**This work is more of an engineering effort. Although the results reported in the paper appear to show some improvement, the methodological innovation and contribution to the community seem limited.**

**Strengths:**

- By combining three metrics (i.e., Disagreement, Entropy, Variance) for few-shot exemplar selection, the method leverages their complementary strengths to improve few-shot prompting performance.
- Incorporating structured entity–relation context has been shown to enhance the performance of CoT, as demonstrated in this paper, and prior studies.
- The presentation is clear, making it easy to read and understand.

**Weaknesses:**

The proposed method consists of two key modules: **uncertainty-driven exemplar selection** and **inference with structural entity-relation context**. However, both modules raise concerns about the novelty and experiments.

[**uncertainty-driven exemplar selection**]
- The proposed metric $\alpha\(x_i\)$ for uncertainty-driven exemplar selection is a simple combination of three metrics. And the three metrics, e.g., entropy [1], have already been leveraged in few-shot exemplar selection, making it a A+B-style work.
- The paper lacks ablation studies on $\lambda_1$, $\lambda_2$ and $\lambda_3$ (Eq.5). It is unclear how these three hyperparameters were chosen (the authors do not even provide the values of the three hyperparameters) and which metric plays the dominant role for $\alpha\(x_i\)$.

[**inference with structural entity-relation context**]
- ERA-CoT [2] have proposed to leverage the entity-relation information to enhance the CoT reasoning, which is dramatically similar to this paper. However, the authors do not explicitly cite this paper when describing their method in Sec.3.2. They just mention it in related works, but do not provide compelling evidence regarding the distinction between the two. Thus, considering ERA-CoT, the novelty of this work in leveraging entity–relation information to enhance CoT reasoning seems also limited.

[Experiments]
- This paper lacks the results of some import baselines. For instance, many other metric-based exemplar selection methods, e.g., MaxEntropy, MaxIG, CBS MaxIG [1],  Similar-ICL [3], self-consistency-based methods, etc.  And methods that enhance CoT reasoning with entity-relation context, especially ERA-CoT.
- This paper lacks the ablation studies about the three metrics. Fine-grained ablation study about how the three metrics impact the final prediction is important and necessary.
- The analyses of the experimental results are somewhat simplistic and superficial, and does not provide any new insights. I may overlook some key points, and I would welcome the authors’ response on this matter during the rebuttal period.

## Reference
[1] Towards Informative Few-Shot Prompt with Maximum Information Gain for In-Context Learning (Liu & Wang, EMNLP Findings 2023)
[2] ERA-CoT: Improving Chain-of-Thought through Entity Relationship Analysis (Liu et al., ACL 2024)
[3] In-Context Learning with Iterative Demonstration Selection (Qin et al., EMNLP Findings 2024)

**Questions:**

N/A

---

### Official Review · Reviewer_J5Rm · 2025-11-01

**Soundness:** 1
**Presentation:** 2
**Contribution:** 2
**Rating:** 2
**Confidence:** 3

**Summary:**

This paper introduces Structure-driven Dynamic Active Chain-of-Thought (SDA-CoT) to address a key weakness of standard CoT: reliance on fixed human exemplars that overlook entity–relation structure and can yield confident but incorrect reasoning. SDA-CoT combines Bayesian Active Learning to select high-uncertainty, high-value exemplars with structure-aware reasoning that extracts entities and relations to build relational scaffolds, guiding logically coherent, context-consistent chains.

**Strengths:**

1. The paper is well-written and easy to follow.
2. Extensive experiments and ablations were conducted on a wide range of datasets.

**Weaknesses:**

1. My biggest concern is that the evaluation metrics in the Main Results section seem incorrect. For example, DeepSeek-R1’s reported accuracy on GSM8K is only 49.0, far below its expected level, which undermines my confidence in the paper’s experimental findings.
2. The claim that uncertain exemplars are more beneficial for CoT reasoning is somewhat counterintuitive; I believe more detailed analysis and evidence are needed.
3. In Figure 1, the authors imply that the computational logic in the exemplars interferes with the LLM’s reasoning, but the analysis lacks rigor. I’m uncertain whether a causal relationship exists or how severe the issue is.

**Questions:**

1. I couldn’t find an explanation of how the questions in $D_{train}$ are constructed. If I’ve overlooked any details, please correct me.
2. Reasoning models like DeepSeek-R1 typically don’t use few-shot CoT exemplars. Given its long reasoning traces, how are the few-shot exemplars constructed for DeepSeek-R1, and do such lengthy exemplars risk exceeding the context-length limit?

---

### Note · Authors · 2025-11-12

I have read and agree with the venue's withdrawal policy on behalf of myself and my co-authors.